

# The crouching of the shrew: Mechanical consequences of limb posture in small mammals

Daniel K. Riskin[1,2], Corinne J. Kendall[1,3] and John W. Hermanson[1]

[1] Department of Biomedical Sciences, College of Veterinary Medicine, Cornell University, Ithaca NY, United States
[2] Current affiliation: Department of Biology, University of Toronto Missisauga, Mississauga, Ontario, Canada
[3] Current affiliation: North Carolina Zoo, Asheboro, NC, United States

## ABSTRACT

An important trend in the early evolution of mammals was the shift from a sprawling stance, whereby the legs are held in a more abducted position, to a parasagittal one, in which the legs extend more downward. After that transition, many mammals shifted from a crouching stance to a more upright one. It is hypothesized that one consequence of these transitions was a decrease in the total mechanical power required for locomotion, because side-to-side accelerations of the body have become smaller, and thus less costly with changes in limb orientation. To test this hypothesis we compared the kinetics of locomotion in two mammals of body size close to those of early mammals ($< 40$ g), both with parasagittally oriented limbs: a crouching shrew (*Blarina brevicauda*; 5 animals, 17 trials) and a more upright vole (*Microtus pennsylvanicus*; 4 animals, 22 trials). As predicted, voles used less mechanical power per unit body mass to perform steady locomotion than shrews did ($P = 0.03$). However, while lateral forces were indeed smaller in voles ($15.6 \pm 2.0\%$ body weight) than in shrews ($26.4 \pm 10.9\%$; $P = 0.046$), the power used to move the body from side-to-side was negligible, making up less than 5% of total power in both shrews and voles. The most power consumed for both species was that used to accelerate the body in the direction of travel, and this was much larger for shrews than for voles ($P = 0.01$). We conclude that side-to-side accelerations are negligible for small mammals–whether crouching or more upright–compared to their sprawling ancestors, and that a more upright posture further decreases the cost of locomotion compared to crouching by helping to maintain the body's momentum in the direction of travel.

Corresponding author
Daniel K. Riskin,
dan.riskin@utoronto.ca

## INTRODUCTION

A key transition in the evolution of mammals was from a sprawling, lizard-like posture to one where the limbs extended ventrally from the body (*Carroll*, *1988*; *Kielan-Jaworowska & Hurum*, *2006*). Further, that transition can be broken down into two biomechanically relevant parts: the first was for the limbs to be adducted, or brought under the body—a character state that persists in most mammals today, and typically sets them apart from quadrupedal vertebrates that sprawl, such as salamanders, tortoises, crocodilians, and

squamates (*Reilly et al., 2006*; *Gatesy, 1991*; *Zani, Gottschall & Kram, 2005*). The second component of that evolutionary transition was for the limbs to be extended below the body, by increasing the joint angles of the shoulder and elbow for example, raising the center of mass (COM), and taking the animal from a crouched posture to an upright one. This second shift was not so universally widespread among mammalian lineages; indeed there is considerable variation among quadrupedal mammals in body size and limb positioning (*Biewener, 2005*; *Fish et al., 2001*; *Parchman, Reilly & Biknevicius, 2003*). Here, we use that diversity to test hypotheses about the mechanical consequences of posture.

It is widely believed that the parasagittal posture of cursorial mammals gives them improved mechanical efficiency while standing and crawling compared with the ancestral, sprawling one of proto-mammals (e.g., *Reilly, McElroy & Biknevicius, 2007*). One key consequence of that evolution has been changes in the magnitudes of laterally directed ground reaction forces. For example, the magnitudes of lateral forces (scaled to body size) are generally smaller for mammals than they are for sprawling animals such as reptiles (*Farley & Ko, 1997*; *Reilly & Blob, 2003*; *Lammers & Biknevicius, 2004*; *Willey et al., 2004 Reilly & Elias, 1998*). The physical reason for this may stem from the angles between the feet and the center of mass in a transverse plane. For a ground reaction force of some fixed magnitude directed toward the animal's COM (*Chen et al., 2006*; *Lee, Bertram & Todhunter, 1999*), the magnitude of the horizontal component will be higher for a sprawling posture than a parasagittal one. Presumably, because the limbs of mammals are less splayed than those of ancestral tetrapods, mammals should spend less energy on laterally oriented ground reaction forces than sprawling quadrupeds do.

The biomechanical consequences of posture along the gradient from crouching to upright are not as clear. We hypothesize that this gradient also influences the magnitudes of laterally oriented forces, by changing the height of the animal's COM. Since the COM of a sprawling animal is closer to the ground than that of an upright walker, the ground reaction forces oriented directly toward the COM should be more horizontally oriented for crouching animals than they are for upright walkers. Based on this assumption, we hypothesize that the mechanical energy required to push the COM from side-to-side (mediolaterally) should be higher for crouching animals than for animals with a more upright posture.

In this study, we compare the locomotion of two small mammal species: a shrew with a crouching posture and a more upright vole. Shrews hold their bodies close to, or in contact with, the ground (*Suzuki, 1990*). Voles carry their bodies off the ground while traveling, and are representative of the more upright condition observed in small mammals. Since we are interested in the evolution of early mammals, which were small, our study does not include large mammals, which possess much more upright postures than any small mammals do (*Biewener, 1989*; *Biewener, 2005*). As such, we do not investigate the full range of postures present among modern mammals, but instead focus on two species that serve as a proxy for the kind of variability that existed when mammals first diversified.

We predict that voles will produce smaller magnitude lateral ground reaction forces, scaled to body weight, than shrews will. We also hypothesize that an upright posture reduces the energetic requirements of terrestrial locomotion by reducing the amount of mechanical

power required to move the animal's COM. We predict that the total mechanical power per kilogram required for locomotion will be higher for shrews than for voles, and that the majority of increased costs for shrews will result from lateral ground reaction forces.

## METHODS

### Force plate trials

We caught five northern short-tailed shrews (Eulipotyphla: *Blarina brevicauda*; body mass ($m_b$) 22.7 ± S.D. 3.8 g) and four meadow voles (Rodentia: *Microtus pennsylvanicus*; $m_b = 29.3 ± 7.8$ g) in live traps near Ithaca, NY. Each animal was used in our experiments within 12 h of trapping and then released at its point of capture. The Institutional Animal Care and Use Committee at Cornell University approved all protocols pertaining to this project, and it was approved by the Cornell Center for Animal Resources and Education (protocol 2003-0055). We used trapping protocols in accordance with the recommendations of the American Society of Mammalogists (*Choate et al.*, *1998*).

We placed each animal in a Plexiglas enclosure (0.48 m length, 0.15 m width, 0.11 m height) that had a force-measuring platform (0.15 m length, 0.15 m width) in the center of its floor. As animals moved back and forth across the platform, we measured ground reaction forces in three dimensions at 1,000 Hz. The plate had a resonant frequency of ≥128 Hz in all directions. We calibrated plates for load response immediately prior to data collection each day, and found linear correlations of force to output voltage within the range of force magnitudes we recorded in our experiments ($r^2 > 0.999$). Drift in the baseline signal was corrected by zeroing signals in each trial with the output voltage of the unloaded plate. The signals from the force plate were post-processed using a 58–62 Hz Butterworth filter to remove AC noise and a Butterworth lowpass filter at 25 Hz to improve the signal to noise ratio overall. Details of plate construction, calibration, and performance are further described elsewhere (*Riskin, Bertram & Hermanson*, *2005*; *Riskin et al.*, *2006*).

While an animal moved across the plate, we recorded video at 250 Hz with a MotionMeter 250 digital high-speed camera (Redlake Systems, San Diego CA, USA). The camera recorded simultaneous lateral and dorsal views of the animal from its position ca. 2 m away, with the assistance of a mirror above the enclosure. Videos were synchronized with force platform recordings to within 0.004 s using the methods of *Riskin, Bertram & Hermanson* (*2005*); *Riskin et al.* (*2006*).

From each trial, we isolated a single stride cycle, beginning and ending with the footfall of the hindlimb facing the camera, where the animal did not make contact with any part of the cage other than the force plate surface. Average speed for the trial was calculated as the horizontal distance traversed by the tip of the nose divided by the period of the stride cycle, and stride frequency was calculated as the inverse of stride period. We discarded trials where the animal stopped and then resumed locomotion mid-sequence, keeping only trials in which the animal kept moving throughout the stride cycle.

### Verification of differences in posture

To visualize the posture of these animals, two other *B. brevicauda* ($m_b$ 18.1 and 22.1 g) and three other *M. pennsylvanicus* ($m_b$ 20.5, 26.0, and 34.9 g) were filmed in lateral view, using a

Philips Easy Diagnost Eleva cineflouroscopic system (Philips N.V., Utrecht, Netherlands). Images were collected at 8 Hz while animals crawled and stood. Digitized images were analyzed to confirm that the postures of the two species differed as predicted. We noted selected bone position and joint angles at several stance phases of the step cycle (early, middle or late stance) and during standing.

## Calculations and statistical analyses

We calculated acceleration, velocity, and position of the COM using numerical integration of the ground reaction forces recorded as animals crossed the force plates. Horizontal forces (fore-aft and mediolateral) and vertical force minus the product of mass and the acceleration due to gravity ($g = 9.81$ m s$^{-1}$) were divided by the animal's body mass to arrive at instantaneous acceleration of the COM. Acceleration was integrated to calculate instantaneous velocity, and vertical velocity was integrated to find height of the COM. To obtain constants for integration of acceleration (initial velocity), we used a custom-made program in Matlab 7.2.0 (MathWorks Inc., Natick, MA, USA) to digitize the movement of the nose tip during the video sequence. To arrive at initial velocity estimates, a linear least-squares best-fit line was calculated for fore-aft and mediolateral changes in nose position over the 10 camera frames (0.04 s) prior to the stride cycle. Changes in the pitch of the body prevented using this method for initial vertical velocity, so we calculated an initial velocity that resulted in a change in height of the COM that matched the net change in height of the nose tip in the video of the stride cycle. The constant for integration of vertical velocity (initial height) was assigned a value of 0 m.

We defined the fore-aft direction as parallel to the horizontal component of the nose's change in position from the first to last camera frames of the stride cycle, and the perpendicular horizontal direction as mediolateral. Kinetic energy in the fore-aft direction was calculated using the equation $E_{KF} = 0.5 m_b v_F^2$, where $v_F$ is the fore-aft velocity. Mediolateral kinetic energy ($E_{KL}$) and vertical kinetic energy ($E_{KV}$) were calculated analogously, but using mediolateral velocity and vertical velocity, respectively, instead of $v_F$. Total kinetic energy was the sum of those three components ($E_K = E_{KF} + E_{KL} + E_{KV}$). Gravitational potential energy was calculated as $E_P = mgh$, where $h$ is the height of the COM. Total energy ($E_{TOT}$) was calculated as the sum of $E_K$ and $E_P$.

Power used to increase any component of energy was calculated as the sum of positive increments in that component of energy over the stride cycle divided by stride period. Power used to increase $E_P$ is denoted by $P_{EP}$, and power used to increase $E_K$, $E_{KF}$, $E_{KL}$, $E_{KV}$, and $E_{TOT}$ are denoted by $P_{EK}$, $P_{EKF}$, $P_{EKL}$, $P_{EKV}$, and $P_{ETOT}$, respectively. Body mass-specific power values, were calculated by dividing power by $m_b$.

We used two-tailed $t$-tests of all trials to verify that the speeds of shrew and vole trials were not significantly different. For all other comparisons between species, however, we pooled all the trials of a given individual, and used individual means for statistical testing. Using one-tailed Student's $t$-tests (*Zar*, *1999*), we tested the hypothesis that shrews had larger peak-magnitude forces in the mediolateral axis than voles did. We also used one-tailed $t$-tests to test the hypotheses that each component of body mass-specific power ($P_{EKF}$, $P_{EKL}$, $P_{EKV}$, $P_{EK}$, and $P_{EP}$) would be higher for shrews than for voles.

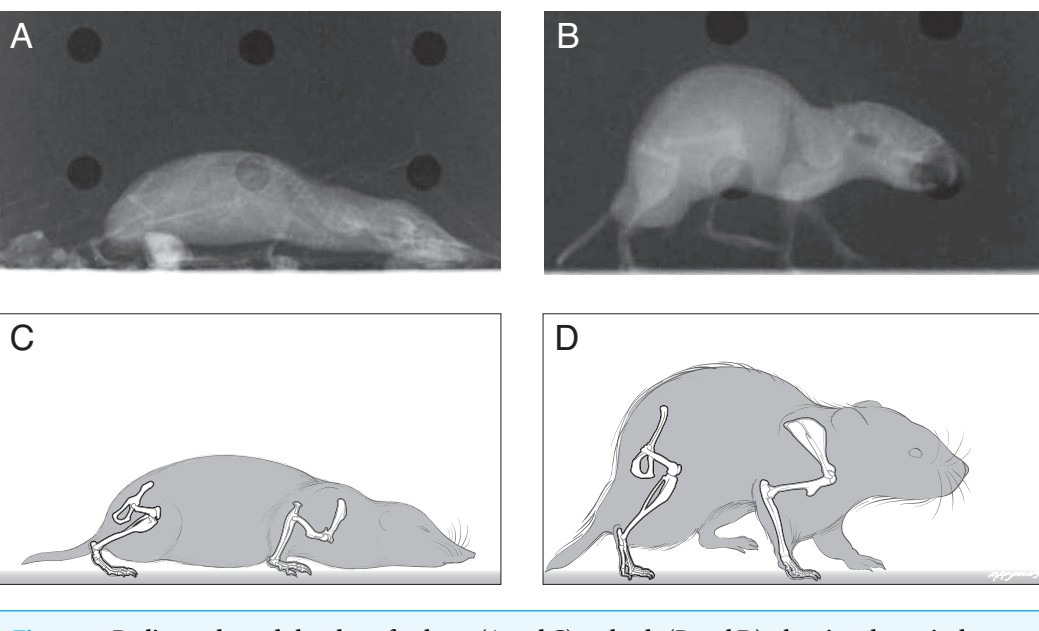

**Figure 1** **Radiographs and sketches of a shrew (A and C) and vole (B and D), showing the typical standing posture of each.** Shrews held their bodies closer to the ground than the voles did.

The influence of speed on stride frequency in each species was assessed using a Standard Least Squares model of all trials with the individual treated as a random effect (*Riskin et al.*, *2010*).

# RESULTS

## Posture

The postures of shrews were more crouched than those of voles. The shrew humerus was often held such that the elbow was dorsal relative to the shoulder joint (Fig. 1) as revealed by an angle of 30° above horizontal (range from −39° at the beginning of stance phase to +39° at middle to late stance phase). This humeral orientation accounted in large part for the crouching posture seen in standing or crawling shrews. The humerus of the voles tended to be held close to or below horizontal throughout the stance phase of locomotion or during quiet standing. There did not appear to be large differences between the shrew and vole in terms of the orientation of the radius/ulna during locomotion.

The shrew femur tended to remain near horizontal (range from −2 to −27°) during locomotion. In contrast the vole's femur spanned a larger range (from 23° above horizontal during early stance to −70°, or well below horizontal during high stance or late in the step cycle).

## Force plate recordings

We recorded 17 trials with shrews traveling at speeds of 0.23 to 0.84 m s$^{-1}$ and 22 trials with voles at speeds of 0.18 to 0.51 m s$^{-1}$ (Table 1). The speeds of shrew trials (mean ± S.D. $= 0.42 \pm 0.15$ m s$^{-1}$, $N = 17$) and vole trials ($0.37 \pm 0.09$ m s$^{-1}$, $N = 22$) did not differ significantly (two-tailed $t = 1.33$; $DF = 23.84$; $P = 0.19$). Both species demonstrated

Riskin et al. (2016), *PeerJ*, DOI 10.7717/peerj.2131

**Table 1** **In this study, animals were recorded with high-speed video and force plates over the course of a stride cycle.** Summary statistics for (A) each of the 39 trials, and (B) averages for each individual's trials are provided here. Note that the full time series of data for each of the 39 force plate trials in (A) is available on the Dryad website.

| species | individual | mass (g) | Forward Velocity (m s$^{-1}$) | Greatest Speed Change (% of average) | Stride Frequency (s$^{-1}$) | Maximum Lateral Force (% Body Weight) | $P_{EKf}$ / mass (watts kg$^{-1}$) | $P_{EKl}$ / mass (watts kg$^{-1}$) | $P_{EKv}$ / mass (watts kg$^{-1}$) | $P_{EK}$ / mass (watts kg$^{-1}$) | $P_{EP}$ / mass (watts kg$^{-1}$) | $P_{ETOT}$ / mass (watts kg$^{-1}$) |
|---|---|---|---|---|---|---|---|---|---|---|---|---|
| (A) **Summary statistics for each of the 39 trials in this study.** | | | | | | | | | | | | |
| *B. brevicauda* | shrew1 | 20.9 | 0.23 | 13.3% | 6.58 | 16.9% | 0.130 | 0.016 | 0.008 | 0.139 | 0.067 | 0.162 |
| | | | 0.33 | 16.7% | 6.41 | 30.5% | 0.025 | 0.025 | 0.006 | 0.021 | 0.036 | 0.019 |
| | | | 0.52 | 28.2% | 7.58 | 24.3% | 1.129 | 0.016 | 0.043 | 1.158 | 0.250 | 1.230 |
| | | | 0.32 | 15.6% | 5.95 | 18.5% | 0.048 | 0.019 | 0.023 | 0.042 | 0.295 | 0.144 |
| | | | 0.35 | 8.4% | 6.76 | 31.5% | 0.175 | 0.028 | 0.037 | 0.192 | 0.172 | 0.337 |
| | shrewA | 27.0 | 0.33 | 24.6% | 3.79 | 15.9% | 0.208 | 0.006 | 0.044 | 0.216 | 0.277 | 0.374 |
| | | | 0.45 | 10.4% | 5.81 | 17.4% | 0.289 | 0.006 | 0.085 | 0.304 | 0.277 | 0.478 |
| | shrewB | 19.2 | 0.25 | 27.4% | 4.24 | 16.9% | 0.131 | 0.006 | 0.026 | 0.131 | 0.150 | 0.256 |
| | | | 0.47 | 4.8% | 6.10 | 10.7% | 0.157 | 0.001 | 0.038 | 0.151 | 0.174 | 0.301 |
| | | | 0.50 | 12.3% | 7.81 | 23.6% | 0.354 | 0.006 | 0.045 | 0.333 | 0.169 | 0.401 |
| | shrewC | 20.0 | 0.57 | 10.6% | 7.35 | 38.5% | 0.493 | 0.012 | 0.062 | 0.468 | 0.229 | 0.650 |
| | | | 0.51 | 12.8% | 6.41 | 43.6% | 0.398 | 0.020 | 0.055 | 0.414 | 0.189 | 0.515 |
| | | | 0.84 | 6.6% | 10.42 | 46.1% | 0.890 | 0.031 | 0.068 | 0.850 | 0.278 | 0.875 |
| | shrewE | 26.6 | 0.36 | 8.2% | 6.25 | 27.6% | 0.151 | 0.009 | 0.018 | 0.150 | 0.090 | 0.227 |
| | | | 0.41 | 20.4% | 5.68 | 20.1% | 0.285 | 0.010 | 0.035 | 0.290 | 0.153 | 0.292 |
| | | | 0.23 | 30.8% | 5.10 | 36.7% | 0.111 | 0.021 | 0.013 | 0.117 | 0.037 | 0.073 |
| | | | 0.49 | 24.3% | 6.41 | 40.2% | 0.602 | 0.006 | 0.023 | 0.609 | 0.174 | 0.714 |
| *M. pennsylvanicus* | vole1 | 23.8 | 0.51 | 28.1% | 6.41 | 25.8% | 0.054 | 0.015 | 0.098 | 0.042 | 0.266 | 0.246 |
| | | | 0.50 | 28.2% | 6.10 | 9.4% | 0.111 | 0.003 | 0.106 | 0.064 | 0.297 | 0.301 |
| | | | 0.35 | 17.0% | 4.17 | 8.8% | 0.220 | 0.001 | 0.080 | 0.222 | 0.276 | 0.471 |
| | | | 0.39 | 25.4% | 5.43 | 11.6% | 0.073 | 0.004 | 0.429 | 0.462 | 0.392 | 0.320 |
| | | | 0.41 | 13.3% | 4.31 | 19.9% | 0.286 | 0.007 | 0.060 | 0.245 | 0.232 | 0.450 |
| | | | 0.34 | 39.3% | 5.00 | 20.0% | 0.052 | 0.008 | 0.132 | 0.059 | 0.380 | 0.360 |
| | | | 0.28 | 11.0% | 4.24 | 13.2% | 0.097 | 0.002 | 0.058 | 0.126 | 0.247 | 0.334 |

Riskin et al. (2016), *PeerJ*, DOI 10.7717/peerj.2131

**Table 1** (*continued*)

| species | individual | mass (g) | Forward Velocity (m s⁻¹) | Greatest Speed Change (% of average) | Stride Frequency (s⁻¹) | Maximum Lateral Force (% Body Weight) | $P_{EKF}$/ mass (watts kg⁻¹) | $P_{EKL}$/ mass (watts kg⁻¹) | $P_{EKV}$/ mass (watts kg⁻¹) | $P_{EK}$/ mass (watts kg⁻¹) | $P_{EP}$/ mass (watts kg⁻¹) | $P_{ETOT}$/ mass (watts kg⁻¹) |
|---|---|---|---|---|---|---|---|---|---|---|---|---|
| | voleA | 39.6 | 0.37 | 25.7% | 4.39 | 15.8% | 0.032 | 0.004 | 0.007 | 0.035 | 0.114 | 0.122 |
| | | | 0.36 | 11.2% | 3.52 | 14.1% | 0.104 | 0.003 | 0.010 | 0.104 | 0.053 | 0.122 |
| | voleB | 31.1 | 0.26 | 23.3% | 3.21 | 16.5% | 0.044 | 0.002 | 0.026 | 0.043 | 0.145 | 0.162 |
| | | | 0.22 | 17.2% | 2.84 | 14.7% | 0.080 | 0.002 | 0.040 | 0.114 | 0.197 | 0.232 |
| | | | 0.18 | 26.3% | 2.72 | 22.8% | 0.042 | 0.005 | 0.012 | 0.045 | 0.143 | 0.165 |
| | | | 0.35 | 21.7% | 3.79 | 16.7% | 0.206 | 0.007 | 0.042 | 0.208 | 0.338 | 0.418 |
| | | | 0.34 | 13.2% | 4.03 | 19.0% | 0.016 | 0.014 | 0.037 | 0.041 | 0.209 | 0.188 |
| | | | 0.33 | 39.6% | 5.21 | 21.1% | 0.047 | 0.005 | 0.092 | 0.069 | 0.236 | 0.259 |
| | voleC | 22.6 | 0.32 | 9.5% | 4.10 | 12.7% | 0.060 | 0.002 | 0.017 | 0.059 | 0.150 | 0.185 |
| | | | 0.33 | 20.6% | 5.00 | 13.0% | 0.033 | 0.003 | 0.127 | 0.062 | 0.200 | 0.186 |
| | | | 0.44 | 9.0% | 5.00 | 5.4% | 0.146 | 0.000 | 0.130 | 0.168 | 0.249 | 0.364 |
| | | | 0.48 | 18.4% | 5.95 | 16.7% | 0.029 | 0.010 | 0.112 | 0.106 | 0.235 | 0.176 |
| | | | 0.41 | 20.3% | 5.81 | 19.0% | 0.007 | 0.002 | 0.085 | 0.043 | 0.291 | 0.176 |
| | | | 0.39 | 16.2% | 4.72 | 16.8% | 0.168 | 0.005 | 0.125 | 0.215 | 0.270 | 0.398 |
| | | | 0.48 | 4.3% | 5.95 | 12.3% | 0.104 | 0.007 | 0.108 | 0.122 | 0.230 | 0.315 |
| **(B) Mean summary statistics for each of the nine individuals in this study.** | | | | | | | | | | | | |
| *B. brevicauda* | shrew1 | 20.9 | 0.35 | 16.4% | 6.65 | 24.3% | 0.302 | 0.021 | 0.023 | 0.311 | 0.164 | 0.378 |
| | shrewA | 27.0 | 0.39 | 17.5% | 4.80 | 16.7% | 0.249 | 0.006 | 0.065 | 0.260 | 0.277 | 0.426 |
| | shrewB | 19.2 | 0.41 | 14.9% | 6.05 | 17.0% | 0.214 | 0.004 | 0.037 | 0.205 | 0.164 | 0.319 |
| | shrewC | 20.0 | 0.64 | 10.0% | 8.06 | 42.7% | 0.594 | 0.021 | 0.062 | 0.577 | 0.232 | 0.680 |
| | shrewE | 26.6 | 0.37 | 20.9% | 5.86 | 31.2% | 0.288 | 0.011 | 0.022 | 0.291 | 0.114 | 0.326 |
| *M. pennsylvanicus* | vole1 | 23.8 | 0.40 | 23.2% | 5.09 | 15.5% | 0.127 | 0.006 | 0.137 | 0.174 | 0.299 | 0.354 |
| | voleA | 39.6 | 0.36 | 18.4% | 3.95 | 14.9% | 0.068 | 0.003 | 0.008 | 0.069 | 0.084 | 0.122 |
| | voleB | 31.1 | 0.28 | 23.6% | 3.63 | 18.4% | 0.073 | 0.006 | 0.042 | 0.087 | 0.211 | 0.237 |
| | voleC | 22.6 | 0.41 | 14.0% | 5.22 | 13.7% | 0.078 | 0.004 | 0.101 | 0.111 | 0.232 | 0.257 |

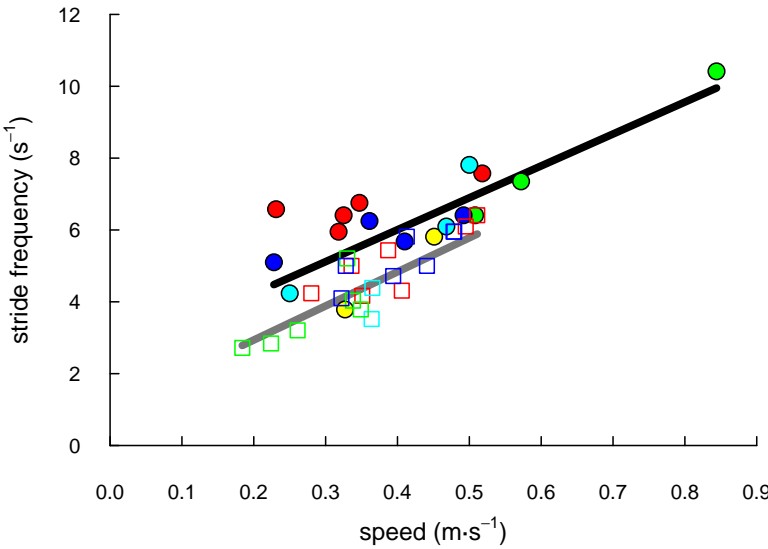

**Figure 2** **The range of speeds over which shrews (solid circles & black line) and voles (open squares & grey line) moved in this study were not significantly different ($P = 0.19$).** Stride frequency increased linearly for both species ($P \leq 0.0002$). Individual animals (5 shrews and 4 voles) are differentiated by marker colour.

a linear increase of stride frequency with speed (shrews: $F(5,11) = 13.89$; $P = 0.0002$; $r^2 = 0.86$; voles: $F(4,17) = 17.41$; $P < 0.0001$; $r^2 = 0.80$; Fig. 2).

As predicted, peak lateral forces were larger for shrews ($26.4 \pm 10.9\%$ body weight, $N = 5$) than for voles ($15.6 \pm 2.0\%$; $N = 4$; one-tailed $t = 2.16$, $DF = 4.34$; $P = 0.046$). Total mechanical power per unit mass ($P_{ETOT}/m_b$) used by shrews during locomotion was greater than that used by voles (one-tailed $t = 2.24$; $DF = 6.78$; $P = 0.03$; Fig. 3). Power used to change kinetic energy ($P_{EK}/m_b$) was greater in shrews than in voles (one-tailed $t = 3.18$; $DF = 4.97$; $P = 0.01$; Fig. 3). This was not simply the consequence of differences in net acceleration, as net change in speed over the stride cycle did not differ significantly between the two species (shrews: $16.0 \pm 3.9\%$, voles: $19.8 \pm 4.5\%$; two-tailed $t = 1.34$, $DF = 6.14$, $P = 0.89$). $P_{EKF}/m_b$ and $P_{EKL}/m_b$ were both significantly higher for shrews than for voles (one-tailed $t = 3.50$; $DF = 4.33$, $P = 0.01$; and one-tailed $t = 2.23$; $DF = 4.22$, $P = 0.04$, respectively; Fig. 3). $P_{EKV}/m_b$ was not significantly different between species (one-tailed $t = -0.99$; $DF = 3.60$, $P = 0.81$; Fig. 3), and neither was $P_{EP}/m_b$ (one-tailed $t = -0.31$; $DF = 5.28$, $P = 0.62$; Fig. 3). Representative force, velocity, and energy profiles over the course of a single stride cycle are shown in Fig. 4.

## DISCUSSION

Compared with crouching shrews, more upright voles used less positive power to increase the kinetic energy of the body during locomotion, making the total requirement of mechanical power per kg for locomotion smaller in voles than in shrews. This suggests that the metabolic cost of locomotion is reduced by an upright gait compared with a crouching one. Our findings therefore support the hypothesis that the shift in mammalian posture, from crouching to upright, was driven by selection for economical locomotion.

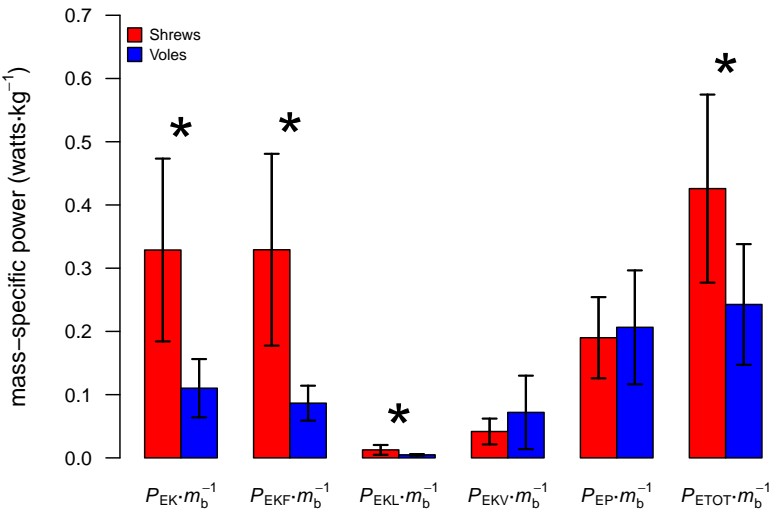

**Figure 3** **Comparison of body mass-specific power used by shrews and voles to increase energy over the course of a stride cycle (mean ± 1 S.D.).** Asterisk denotes a significant difference ($P < 0.05$). The increased mass-specific power requirements of locomotion for shrews result mostly from the relatively higher cost of acceleration in the direction of travel ($P_{EKF}$), not from the power needed to move the body mediolaterally ($P_{EKL}$).

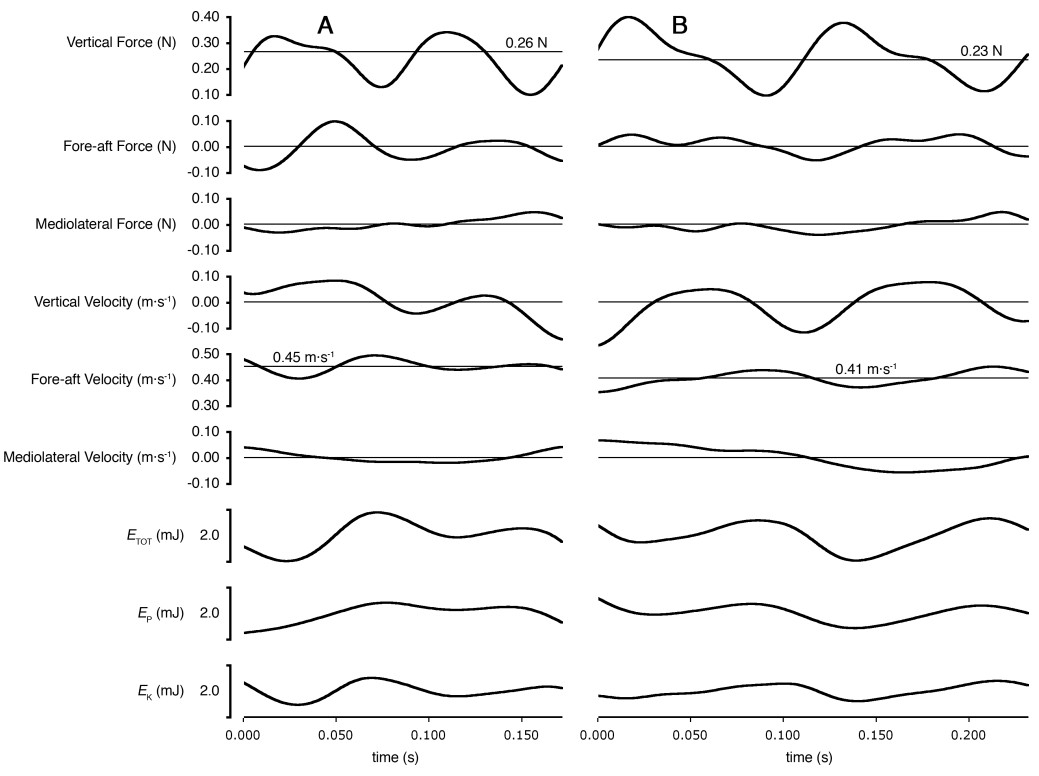

**Figure 4** **Ground reaction forces, COM velocities, and COM kinetics during a stride cycle for (A) a Short-tailed Shrew (mass = 27.0 g; mean forward velocity = 0.45 m s$^{-1}$) and (B) a Meadow Vole (23.8 g; 0.41 m s$^{-1}$).** Thick black lines are experimental data. Thin lines represent zero on most plots, except Vertical Force and Fore-aft velocity, where thin lines represent body weight and mean forward velocity, respectively.

However, we had predicted that the energy spent moving the COM side-to-side in the mediolateral direction would be higher for shrews than voles, and that this would underlie the increased mechanical cost of locomotion while crouching. Indeed, while lateral forces were larger for shrews than for more-upright voles, this was not the major contributor to differences in overall power requirements between species; the magnitude of power used to move the body in the mediolateral direction ($P_{EKL}$) was less than 5% of the total power used to accelerated the center of mass ($P_{EK}$) in both species. This is remarkable, considering that lateral ground reaction forces can exceed fore-aft forces in sprawling animals, such as running geckos, for example (*Chen et al.*, *2006*). Our results therefore suggest that laterally oriented ground reaction forces diminished in magnitude with the transition from a sprawling to a parasagittal stance in mammals, and that they are small for mammals, regardless of how upright they stand on their parasagittal limbs.

Voles used less mechanical power during locomotion than shrews did because they maintained momentum in the direction of travel, and thus used less energy to accelerate and decelerate their bodies in that direction than shrews did. Power used to accelerate the center of mass in the direction of travel ($P_{EKF}$) was larger for shrews than for voles, and made up more than 80% of $P_{EK}$ for both species. Importantly, this was not the result of speed changes over the courses of trials, because those were roughly equivalent between species. These results support the hypothesis that the evolution from a crouching to upright stance in early mammals was driven by selection for economical locomotion, and that reduction of forces in the craniocaudal axis is the primary mechanism of energy savings.

The mass-specific cost of transport is higher for small terrestrial mammals than for large ones (*Biewener*, *2005*), making the mammals in this study appropriate proxies for early mammals. Importantly, though, our perspective of mammalian postures along a gradient from crouching to upright neglects much of the diversity of limb postures present in mammals, especially the upright postures of large mammals (*Biewener*, *2005*; *Jenkins*, *1971*; *Fischer & Blickhan*, *2006*). Our understanding of mammalian locomotory energetics will be greatly improved by future cineradiographic studies of limb kinematics of mammals with broad ranges of limb postures (e.g., the XROMM method of *Brainerd et al.*, *2010*).

Both species studied here used gaits that kept at least one foot in contact with the ground at all times, and neither switched to kinematically distinguishable gaits at high speeds. These observations are consistent with observations of other small-bodied mammals ($<2$ kg) by *Biknevicus et al.* (*2013*), who postulated that the limited utility of elastic energy recovery from tendons for small mammals, compared with large mammals, makes bouncing gaits less worthwhile. However, *Schilling & Hackert* (*2006*) recorded bounding and half-bound gaits among metatherian and eutherian mammals weighing as little as 145–200 g, and demonstrated that flexion of the vertebral columns effectively increased stride length in those running gaits, beyond what is possible for walking alone.

The persistence of crouching in shrews, despite the mechanical advantages of a more upright gait over a crouching one, suggests that some other selective benefits to crouching exist. Ease of movement through tight burrows and reduced conspicuousness to predators are two obvious ones. Another would be readiness to accelerate quickly, without the need for a pre-jump crouch that would be necessary for an animal standing on outstretched limbs.

Furthermore, *Fischer* (*1994*) has argued that advantages of crouching include an improved ability to maneuver around unexpected obstructions. This fits well with work on guinea fowl showing that their bent limbs help them navigate obstacles during bipedal locomotion (*Daley & Biewener*, *2011*). For small mammals, like those in our study, a terrain that might seem flat to a large upright mammal such as an ungulate might be riddled with obstacles even taller than they are. Indeed, while the mechanics of locomotion over smooth surfaces described here are essential for understanding the basic biomechanics of locomotion for these animals, future studies focused on locomotion across variable terrain are needed to fully understand the energetic significance of posture for small mammals in the wild.

## ACKNOWLEDGEMENTS

The authors thank Allison Devlin for assistance with pilot studies and Drs. Howard E. Evans, Milo E. Richmond, and Richard A. Malecki for assistance locating study animals. Michael A. Simmons, MFA, made the line drawings in Fig. 1. Dr. Kevin M. Middleton and Dr. Rebecca M. Mitchell assisted with statistical procedures, and Dr. Stephen Gatesy and three anonymous reviewers provided critical review of earlier drafts of the manuscript. We thank Anthony DeLaurentiis for conducting the fluoroscopic work. This study was part of an undergraduate Physiology Honors Thesis research project conducted by CJK at Cornell University.

### Funding

The authors received no funding for this work.

### Competing Interests

The authors declare there are no competing interests.

### Author Contributions

- Daniel K. Riskin conceived and designed the experiments, performed the experiments, analyzed the data, contributed reagents/materials/analysis tools, wrote the paper, prepared figures and/or tables, reviewed drafts of the paper.
- Corinne J. Kendall conceived and designed the experiments, performed the experiments, analyzed the data, wrote the paper, reviewed drafts of the paper.
- John W. Hermanson conceived and designed the experiments, performed the experiments, analyzed the data, contributed reagents/materials/analysis tools, wrote the paper, reviewed drafts of the paper.

### Animal Ethics

The following information was supplied relating to ethical approvals (i.e., approving body and any reference numbers):

The Institutional Animal Care and Use Committee at Cornell University approved all protocols pertaining to this project and it was approved by the Cornell Center for Animal Resources and Education (protocol 2003-0055).

## Data Availability

Dryad: https:dx.doi.org/10.5061/dryad.13ks6.

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
