# Peer review of "The crouching of the shrew: Mechanical consequences of limb posture in small mammals"

_PeerJ, doi:10.7717/peerj.2131_

## Round 0.1 · original submission · Major Revisions

The reviewers have some very constructive comments on the paper that will improve it when implemented. I agree 100% that the revised MS must be crystal-clear about the mediolateral (sprawling vs. parasagittal) vs. vertical (crouched vs. upright) distinctions. Please be sure to include a point-by-point Response to the reviewers. I will decide then whether to re-review or not. It is "moderate revisions" more than major/minor. Thanks for submitting this interesting paper!

Reviewer 1 ·

Basic reporting

See general comments

Experimental design

See general comments

Validity of the findings

See general comments

Additional comments

Line 80-81: “side-to-side’ suggests lateral…..I think you need to reword here to better state that you mean fore-aft (i.e. crouched posture required more fore-aft force because the GRF vector is more horizontally oriented)

84-85: I find it very odd that a vole can be classified as having an upright posture. I think the authors really need to give some definitions, based on quantitive metrics, of what postures defines crouched vs. upright…..there are probably such data on a handful of species in the literature. By compiling such data, perhaps they could better show that the vole is indeed ‘upright’ or that it is actually crouched, just less so (as compared to an ungulate which is truly upright).

94-95: so based on what I think your argument is saying at lines 80-81, you might also include fore-aft forces as part of this cost.

122: ‘walked’ were the animals walking or running? Better to say, moved across the plate. Walking and running have specific definitions. You actually have all the data you would need to compute the phase shift of the KE and PE and also percent recovery from pendular mechanics. Based on your sample traces, it looks like KE and PE are in-phase and thus the animals are running. Also, what about comparing the magnitude of the KE and PE to get a sense of whether these animals are more cursorial or more lumbering? See: Biknevicius, A.R., Reilly, S.M., McElroy, E.J., and Bennett, M.B. 2013. Symmetrical gaits and center of mass mechanics in small bodied, primitive mammals. Zoology. 116:67-74 AND Reilly, S.M., E.J. McElroy, R.A. Odum, V.A. Hornyak. 2006. Tuataras and salamanders show that walking and running mechanics are ancient features of tetrapod locomotion. Proceedings of Royal Society of London B. 273: 1563-1568

190-203: It seems like there should be more details about the data and statistical comparisons. Means and S.D? Statistical comparisons of these means? I think you need this to say the postures are (or are not) different.

133: Was the 50% window needed because these animals rarely moved at steady speed? Is this a key feature of their locomotion? I understand the need to delineate the used trials due to the steady speed assumptions of some of the biomechanics analyses…..however, the reality for these animals might be that they rarely move at a steady speed……so why not include these as part of your data (i.e. we ran the animals xxxx times and only xxx trials fit our criteria for steady speed.

209: report DF for the t-test

201: report numerator and denominator DF for the linear models.

212-222: there are specific conditions for using 1 vs. 2 tailed test. You need to justify the use of a one-tailed test if you are going to use it! This is particularly relevant for your results, as most one-tailed p-values you report would not be significant if they were two tailed p-values.

Results: could any of the mechanical data (force, energy, power) be impacted by speed. I realize that the t-test showed that the two species ran similar speeds….however, there may still be a speed effect on total mechanical power and if so, you might expect that shrews would show larger force, energy, power because they ran slightly (albeit non-significantly) faster. Can you also correct for speed like you corrected for mass, such that power is in watts per kilogram-meter (adjusted for speed and mass)

224: Again, I am not convinced that you can really call a vole ‘upright’, a least not like you would refer to a horse as upright. I think you are likely really looking a variation within a range of crouched postures.

231: side-to-side instead of ‘back and forth’

238-239: Reword the sentence beginning “These results” I would avoid saying ‘we overestimated’ and instead more simply say that the lateral forces were less important for the overall mechanical power.

241: again, I am not sold on calling a vole ‘upright’!

260: however, they could be mechanically distinguishable based on phase-shifts between the KE and PE….I suggest adding this analysis.

265: not vaulting, but spring recovery from tendons. Vaulting refers to holding the leg rigid and moving the COM over it …also called walking or pendulum mechanics.

270: what about rapid acceleration? From a crouched posture an animal is ready to ‘jump’ into running (or just jump!) whereas an upright animal typically has to counter-move first. This would give crouched animals an advantage of more rapidly accelerating. I don’t have a reference off the top of my head for this (and there may not be one comparing crouched or sprawling to erect animals)…..but it seems likely and could be posited as an additional hypothesis for the advantage of crouchedness.

Reviewer 2 ·

Basic reporting

Professional English language was used throughout the manuscript. The introduction informs the reader of the context (the postural changes during mammalian evolution). The authors make clear that the current manuscript is not focussed on the shift from a sprawling to a parasagittal limb orientation, but on a secondary trend within mammals towards a more erect limb posture. I like this differentiation, because in the literature this is often lumped together simply as the sprawling-to-erect paradigm and usually the secondary component is not getting enough attention. I was a little confused to see that the authors apparently still link the reduced magnitude of laterally directed ground reaction forces to this secondary component (e.g. line 56-60). To my understanding the change in the magnitude of laterally directed GRFs is linked to the primary shift from sprawling to parasagittal. This is also stated later by the authors (lines 64-69). For example, the authors should consider exchanging the somewhat unclear word “upright” with “parasagittal” in line 56 to clarify. The structure conforms to the PeerJ standard. All figures are relevant and of high quality. Labelling could also include the individual unless better justification for pooling data is provided (see below). Raw data is provided.

Experimental design

The study presents original primary research within the scope of PeerJ. The research question is well defined and meaningful (but see my comment on clarifying the abstract below). The investigation is performed to a high technical and ethical standard. The methods are described with sufficient detail to replicate.

Validity of the findings

For the most part, the data is robust (but see my comment on pooling data below), statistically sound and controlled. The conclusion is linked to the original research question. All speculation is identified as such.

Additional comments

Abstract: Please consider to re-word lines 18-24 to clarify that here not the consequences of a change from sprawling to parasagittal limb orientation are studied, but from crouched to erect. In a crouched posture the limbs are not oriented downward (line 20) and a crouched posture is not upright. This should be disentangled.
Line 56: Please consider exchanging “modern” with “cursorial”.
Line 66: Note that experimental data on sprawling animals show that the GRF not always points towards the animals CoM, but instead usually the medio-lateral force is <10° from vertical (cf. e.g. Chen et al 2006 JEB; Kawano and Blob 2013 Int Comp Biol, Nyakatura et al 2014 Evol Biol).
Line 69: The sentence starting with “Because…” is presented as a fact. There should be a citation.
Lines 128-135: Justification for pooling data. Given the relatively small sample size (17 and 22 trials, respectively) what was the justification for pooling the data for each species? How many trials of each individual were used? Did the individuals of a species differ significantly from each other? I suppose this was not the case, but was the sample size even big enough to even test this? A table presenting these data would help the reader to better understand the data basis of the study.
Line 134: Instantaneous deviation of up to 50% from average speed was accepted to include trials for further evaluation. This appears to be quite far from an ideal steady-state trial. Deviation from steady state should have enormous impact on center of mass energetics and hence locomotor costs. How much did the individual trials depart from perfect steady-state? Again, a table would help to better understand the data basis of the study.
Line 138: A different set of animals was used to “verify” the difference of posture. How did these animals compare to the ones used for the GRF measurements? Were they of the same size and mass?
Line 142: Why not show a fluoroscopic image to visualize the difference in posture? I believe an objective image will be more convincing than the line drawing (fig. 1).
Line 168: The sentence starting with “Mediolateral…” is unclear. I get it, but certainly mediolateral kinetic energy was not calculated in the same manner as fore-aft kinetic energy (you need mediolateral velocity).
Line 184: Where are these data? What is the mean of each individual?
Line 193: Please provide more precise data. What was the difference in (mean) humeral and femoral orientation at a) touch down, b) mid-contact, and c) lift off? At what speeds were these measurements made? What are the differences between individuals? This would allow the reader to judge about the posture between both species.
Line 210: This result is probably just an artefact of limited sample size. In most animals studied so far stride frequency converges to a certain value.
Line 228: This result/interpretation really hinges on the assumption of a steady-state gait in both species (see above).

---

## Round 0.2 · accepted · Accept

These are superb revisions and a model Rebuttal document. It made it so easy for me to accept this without further review- the MS has improved a lot and the justification for not including %Recovery (which many have doubts about anyway as a valid metric) is sound; as is the omittance(?it's early here) of EMA (much as that would be nice to have). Oh and thanks for including your forceplate data- that is excellent; others may find it useful/interesting.

The paper should be out soon- congrats!!

One thing: the review process was so well handled on both sides here that I strongly urge you to make the full reviews open (i.e. published). However if you wish not to do so, I understand.